# Contact Points Discovery for Soft-Body Manipulations with Differentiable Physics

**Sizhe Li** *
University of Rochester
sli96@u.rochester.edu

**Zhiao Huang** *
UC San Diego
z2huang@eng.ucsd.edu

**Tao Du**
MIT
taodu@csail.mit.edu

**Hao Su**
UC San Diego
haosu@eng.ucsd.edu

**Joshua B. Tenenbaum**
MIT BCS, CBMM, CSAIL
jbt@mit.edu

**Chuang Gan**
MIT-IBM Watson AI Lab
ganchuang@csail.mit.edu

## Abstract

Differentiable physics has recently been shown as a powerful tool for solving soft-body manipulation tasks. However, the differentiable physics solver often gets stuck when the initial contact points of the end effectors are sub-optimal or when performing multi-stage tasks that require contact point switching, which often leads to local minima. To address this challenge, we propose a contact point discovery approach (CPDeform) that guides the stand-alone differentiable physics solver to deform various soft-body plasticines. The key idea of our approach is to integrate optimal transport-based contact points discovery into the differentiable physics solver to overcome the local minima from initial contact points or contact switching. On single-stage tasks, our method can automatically find suitable initial contact points based on transport priorities. On complex multi-stage tasks, we can iteratively switch the contact points of end-effectors based on transport priorities. To evaluate the effectiveness of our method, we introduce PlasticineLab-M that extends the existing differentiable physics benchmark PlasticineLab to seven new challenging multi-stage soft-body manipulation tasks. Extensive experimental results suggest that: 1) on multi-stage tasks that are infeasible for the vanilla differentiable physics solver, our approach discovers contact points that efficiently guide the solver to completion; 2) on tasks where the vanilla solver performs sub-optimally or near-optimally, our contact point discovery method performs better than or on par with the manipulation performance obtained with handcrafted contact points. Demos are available on our project page[1].

## 1 Introduction

Soft body manipulation has a wide application in cooking (Bollini et al., 2013), fabric manipulation (Wu et al., 2020), healthcare (Mayer et al., 2008) and manufacturing of deformable objects (Sanchez et al., 2018). Differentiable physics has recently been shown as a powerful and effective tool for solving control problems for soft-body manipulation tasks. As demonstrated in Huang et al. (2021), given a parameterized manipulation policy, the differentiable physics solver computes the gradients of the policy parameters, enabling gradient-based optimization much more efficiently than reinforcement learning algorithms at finding optimal solutions for soft-body manipulation tasks on a diverse collection of environments.

However, the performance of the stand-alone gradient-based solver can be heavily influenced by the policy initialization. Especially, the end effectors' initial **contact points** with objects play critical roles in the optimization. Different contact points may lead to vast differences in manipulation performance due to local optima. Besides, some tasks require agents to switch contact points during the manipulation, where the local optima issue becomes a serious bottleneck for completing these

---

*Equal Contribution
[1]Project Page: http://cpdeform.csail.mit.edu

multi-stage tasks. For example, as shown in Figure 2, an agent needs to control the capsule "pen" to sculpt two scribbles on the surface of a yellow plasticine cube. In order to complete the second line, the agent needs to switch contact points after drawing the first one. While the stand-alone differentiable physics solver could possibly draw the first line, it often gets stuck and struggles to draw the second one, due to the lack of gradients that push the pen to a new contact point to begin the second line. How to automatically find proper contact points for soft body manipulation tasks remains a challenge in differentiable physics.

In this paper, we propose a principled framework that integrates an optimal transport-based contact discovery method into differentiable physics (CPDeform) to address this important challenge. CPDeform heuristically find contact points for end effectors by using transport priorities computed from optimal transport to compare the current shape with the target shape, where soft-body manipulation is treated as a particle transportation problem. After finding contact points, CPDeform can combine the differentiable physics solver to solve soft body manipulation tasks. On single-stage tasks that do not require contact point switching, CPDeform can find suitable initial contact points to finish the task.

On multi-stage tasks, using an example shown in Figure 1 (right) where the goal is to reshape a plasticine cube into an airplane, CPDeform can iteratively switch the contact points of end effectors based on transport priorities. This iterative deformation process is motivated by how humans manipulate plasticine. As shown in Figure 1 (left), when humans manipulate a plasticine dough, they tend to repeatedly focus on the point of interest and modify it towards the target shape. CPDeform can mimic this process by iteratively switching contact points of interests based on transport priorities and deforming the soft bodies into the target shape with the help of the differentiable solver. By integrating contact points discovery into the differentiable physics solver, CPDeform can skip over the local minima caused by contact switching and improve the performance of the stand-alone solver.

To evaluate the effectiveness of CPDeform, we introduce PlasticineLab-M that extends the existing differentiable physics benchmark PlasticineLab (Huang et al., 2021) to seven new challenging multi-stage soft body tasks. Extensive experimental results suggest that: on single-stage tasks where the vanilla differentiable physics solver performs sub-optimally or near-optimally in PlasticineLab, we find that the backbone of CPDeform, a contact point discovery method based on optimal transport, single-handedly performs better than or on par with the manipulation performance obtained with random-selected or human-defined contact points. On multi-stage tasks that are infeasible for the vanilla gradient-based solver, we find that CPDeform performs reasonably well in practice and the iterative deformation method equipped with contact point discovery could serve as an alternative to the expensive long-horizon searching algorithm. In summary, our work makes the following contributions:

- We perform an in-depth study of local optimum issues of differentiable physics solver for initial contact points and switching contact points.
- We propose a principled framework, CPDeform, that integrates optimal transport-based contact discovery into differentiable physics.
- We find that the backbone of CPDeform, the contact point discovery method, can be directly employed by the stand-alone solver to find better initial contact points for single-stage tasks.
- On multi-stage tasks, which are infeasible for the vanilla solver, CPDeform employs a heuristic searching approach to iteratively complete the tasks.

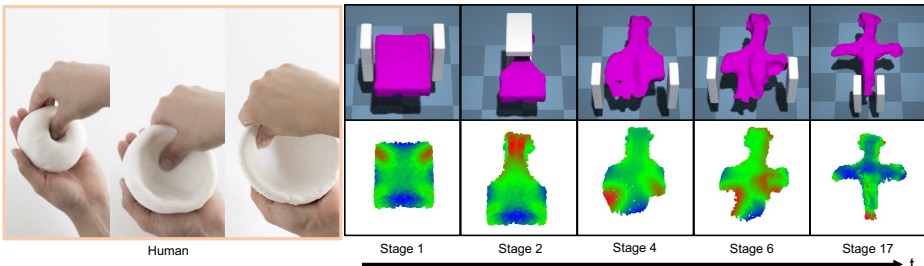

Figure 1: **Left**: the process that a human applies to transform a plasticine dough into a bowl. **Right**: we use images captured in time sequence to demonstrate how our framework reshapes a plasticine cube into a target airplane. The bottom row indicates the transport priorities found by our framework by computing the optimal transport between the current and target shapes. The top row illustrates the pose of the end-effectors selected by our framework based on the transport priorities.

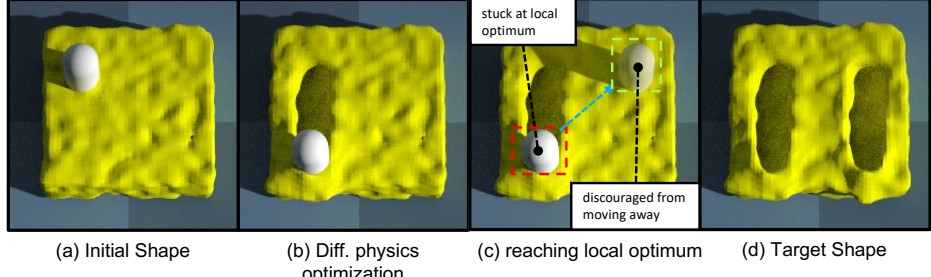

| (a) Initial Shape | (b) Diff. physics optimization | (c) reaching local optimum | (d) Target Shape |

Figure 2: (a) **Writer** to manipulate a "pen"; (b) given an initial handcrafted position of the "pen", the differentiable physics solver can control the "pen" to minimize the loss; (c) the solver gets stuck and fails to finish the second curve, due to the lack of gradient to push the "pen" towards the second line. (d) shows the target shape.

## 2 MOTIVATION

In this section, we provide an intuitive analysis of the drawback of the differentiable physics solver through motivating toy examples. We start with a brief review of how the differentiable physics solver could be employed to optimize manipulation policies. We then demonstrate how initial contact points affect the optimization performance. Finally, we take a simple but representative multi-stage task as an example and discuss why contact switching would often lead to local minima.

We study a **Writer** task as shown in Figure 2(a). In this task, an agent needs to manipulate the capsule "pen" to initiate contact with the yellow plasticine cube, and sculpt a line scribble on the plasticine surface. The agent can move the tip of the pen along three dimensions.

To solve this task with differentiable physics, we manually initialize the end-effector "pen" near the suitable contact point that allow the "pen" to initiate contact with the plasticine. We then parameterize the desired motion trajectory of the "pen" as a sequence of three-dimensional actions $\{a_1, \ldots, a_T\}$ where $T$ is the number of simulation steps. Let $\{s_t\}_{0 \leq t \leq T}$ be simulation states at different time steps which include the state of the plasticine and manipulator. The differentiable simulator starts from the initial state $s_0$ and completes the action sequences by repeatedly executing the forward function $s_{t+1} = \phi(s_t, a_{t+1})$ until the ending state $s_T$ has been reached. The objective of the optimizer is to minimize the distance between the current and target shapes. We represent the objective as a loss function $\mathcal{L}(s_T, g)$ where $g$ is the target shape. Since the simulation forward function $\phi$ is fully-differentiable, we can compute gradients $\partial \mathcal{L}/\partial a_t$ of the loss $\mathcal{L}$ with respect to action $a_t$, and run gradient descent steps to optimize the action sequences by $a_t = a_t - \alpha \partial \mathcal{L}/\partial a_t$ where $\alpha$ is the learning rate. As shown in Figure 2(b), we can see that the agent succeeds at sculpting the target scribble by moving the "pen" downwards. We refer the readers to Algorithm 2 in Appendix D for more details on differentiable physics for controller optimization.

However, if positions of the end-effectors are not well-initialized, the solver would get stuck in the local minima. Taking the task shown in Figure 3 as an example, we illustrate the optimization outcomes with different contact points by showing their corresponding resulting shapes. Even with an arbitrarily large number of steps $T$ given, the gradient-based solver is unable to discover a policy that moves away from the local optimum to a new contact point that allows for task completion. Such phenomena are commonly observed across soft body manipulation tasks using the differentiable physics solver (Huang et al., 2021). When end-effectors are far away from the region of interest, it is often unlikely that the gradient could push the end-effectors towards the desired region. This observation poses the question of how to efficiently find optimal contact points to place the end-effectors.

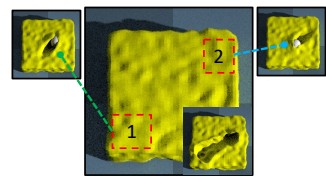

Figure 3: Writer-v1 in PlasticineLab. The bottom-right figure shows the target shape.

The local minima problem caused by inappropriate contact points becomes a more serious issue in multi-stage tasks. Taking the multi-stage writer task in Figure 2(c) as an example, the agent needs to write an additional line on the plasticine surface by switching its contact point. Even with a

well-initialized contact point for the first line, the solver is unable to relocate to the new region of interest for the upcoming line. We observe that, differing from the vanilla differentiable physics solver, humans tend to employ an explicit "iterative deformation" schema to complete such a task. Humans would decompose this task into two stages. In each stage, we tend to iteratively derive the correspondence between the current and target shapes to arrive at useful contact points from observations and then subsequently move the "pen" and write the lines. This motivates us to combine contact point discovery with iterative deformation.

## 3 Method

In this section, we introduce CPDeform, a principled framework that integrates optimal transport-based contact discovery into differentiable physics for solving challenging soft-body manipulation tasks. We first describe our contact point discovery method in relation to the transport priorities found by optimal transport in Section 3.1. In Section 3.2, we describe how transport priorities can be used to place end-effectors. Finally, in Section 3.3, we show how CPDeform integrates contact points discovery with differentiable physics and iteratively deforms the soft bodies for multi-stage tasks.

### 3.1 Optimal Transport and contact point Discovery

One way to consider soft-body manipulation is by treating it as a particle transportation problem. By evaluating the cost of transporting the current state particles $\mu$ to the target state particles $\nu$, optimal transport provides a useful framework for comparing differences between any given pair of shapes, which can guide us to discover contact points. Let all particles be weighted equally in our simulator. Given a cost matrix $M$, optimal transport finds a transportation plan $P$ in transportation polytope $U$ by minimizing the transportation cost $\min_{P \in U} \langle P, M \rangle$. Casting the problem into dual form, we have $\text{OT}(\alpha, \beta) := \max_{f,g} E_\mu[f] + E_\nu[g]$ such that $\forall i, j$, Lagrange multipliers $f_i, g_j$ satisfy $f_i + g_j \leq M_{ij}$, where $\alpha, \beta$ are the mass vectors for the particles in $\mu, \nu$ respectively. We refer the reader to Appendix B for more details on optimal transport. We focus on the Lagrange multipliers $f$ of the source particles, which we refer to as the dual potentials. Since it represents the supports of the source measure, we interpret $f$ as the *transport priorities* for the source particles $\mu$.

Transport priorities are helpful for selecting contact points. Given a pair of current and target shapes, we intuitively would place the end-effectors around the region of the largest difference between the two shapes, in order to substantially modify the shapes. This observation leads us to place the end-effectors at contact points whose corresponding optimal manipulation policies can minimize the shape difference. However, it is computationally prohibitive to directly evaluate the optimality of the contact point by exhaustively searching through a set of contact points. Thus, we propose to heuristically identify contact points, based on a simple rule of selecting contact points with high transport priorities. We observe that contact points with high transport priorities mostly correspond with superior optimization performances.

### 3.2 End Effector Placement With Transport Priorities

In this section, we describe how manipulators are placed at advantageous locations based on optimal transport priorities. We first describe the single manipulator case, which can be extended to multiple manipulator environments by adding a heuristic.

**Direct Single Manipulator Placement.** The transport priorities can be directly employed to place the single manipulator. We identify the source particle $\mu^*$ corresponding to the largest potential $f^*$. We then find a suitable contact point around $\mu^*$ through grid search, where we create a 3D grid centered at $\mu^*$ with each dimension evenly spaced into $N_i$ intervals. The criterion of the grid search is as follows: We reject points whose placements of the manipulator lead to collisions with the particles. And we iterate over each grid point $x \in \mathbb{R}^3$, and place the manipulator at the point with the maximal score according to the following criterion: $\text{score}(x) = \frac{1}{N_p} \sum_{i=1}^{N_p} \frac{f_i}{d(\mu_i, x)^2 + 1}$ where $d(\mu_i, x)$ computes the signed distance between the particle $\mu_i$ to the closest point on the manipulator placed at grid point $x$. As shown in Figure 4 (top), the criterion leads us to discover contact points that allow the manipulator to cover high potential particles while reducing subsequent changes to the low potential particles.

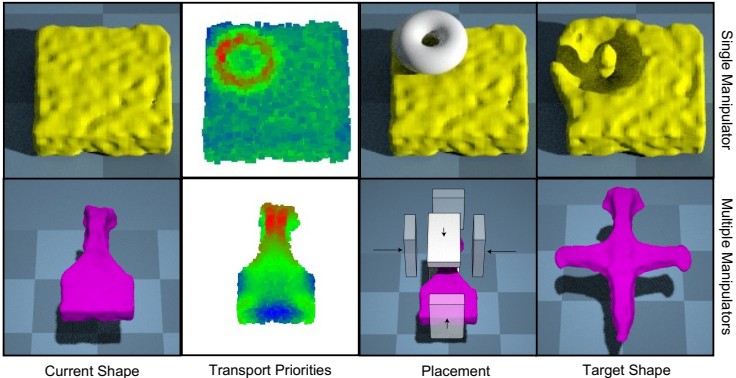

Figure 4: Visualizations of the placement strategies. We visualize the current and target shapes, transport priorities for contact discovery, and the resulting placement of the manipulators. We show a single manipulator case using **Torus** (top), and a multiple-manipulator case using **Airplane** (bottom)

**Multiple Manipulator Placement With Heuristic.** For multiple manipulator environments, we need to consider a heuristically defined candidate pose set $\mathcal{T}$, where each pose corresponds with a different manipulation strategy. Taking Figure 4 (bottom) as an example, it is more advantageous to use the highlighted pose than the ones drawn with low opacity. Each pose is specified by the orientations of the manipulators and the direction vectors for placing other manipulators in relation to the first manipulator. For each pose, we employ the single-manipulator placement strategy for the first manipulator. Using the first placement position as a starting point, for the remaining manipulators, we search along their pose-specific directions to find placement positions with minimal distances from the first manipulator that do not result in collisions with soft bodies. Since the differentiable physics solver is able to effectively adjust manipulator orientations during optimization, in practice, the candidate pose set does not need to be exhaustively large. In practice, we use three poses to cover left-right, top-bottom, and front-back grasping poses.

### 3.3 ITERATIVE DEFORMATION FOR MULTI-STAGE SOFT BODY MANIPULATION

Continuing with the focus on multi-stage soft-body manipulation tasks, we now describe how CPDeform combines the contact point discovery method with the use of differentiable physics for these tasks. For every stage, we iteratively specify the contact points and perform stage-level manipulation with the differentiable physics solver. For the multiple manipulator case, we search over contact plans corresponding to different poses, and choose the plan that achieves the lowest loss.

We present the detailed Algorithm 1 in Appendix C. We start with the initial shape $S_0$. Our goal is to reach the target shape $G$ within a given number of stage $n_{stage}$. Each stage contains $n_{step}$ simulation steps. As described in Section 3.2, we consider a heuristically defined candidate pose set $\mathcal{T}$. In each stage, we first compute the optimal transport between the current shape $S_i$ and the target shape $G$ to derive the transport priorities. Next, we select the particle $p$ with the largest transport priority. For each pose, we search for the placement in relation to $p$ that maximizes the aforementioned criterion.

## 4 EXPERIMENTS

We conduct multiple experiments to test the efficacy of CPDeform on soft-body manipulation tasks to address the following questions:

- On multi-stage tasks that involve multiple contact switches, can CPDeform complete these tasks by iteratively manipulating the soft-bodies?
- How robust is the backbone of CPDeform, the contact point discovery method, when tested on single-stage tasks where it is only allowed to specify a single contact point in one shot?
- How is transport priority compared with other contact discovery methods?

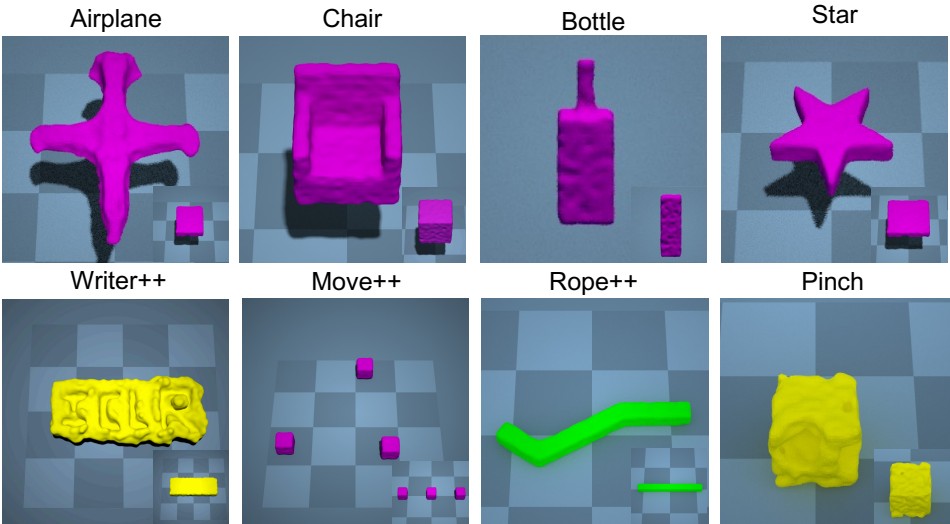

Figure 5: Task illustrations of PlasticineLab-M with target and initial shapes

## 4.1 DATASETS

To extensively evaluate our method, we introduce PlasticineLab-M, a dataset that extends the existing differentiable physics benchmark PlasticineLab with seven new challenging multi-stage soft-body manipulation tasks, and contains the multi-stage environment **Pinch** in PlasticineLab.

We show these eight multi-stage tasks in Figure 5, whose task descriptions are detailed in Appendix A. We also use the remaining single-stage tasks in PlasticineLab to evaluate our contact point discovery method. For multi-stage environments, we use the Wasserstein-1 distance (Solomon et al., 2015) approximated by Sinkhorn iteration (Cuturi, 2013) between the source and target particles to quantify the fine-grained difference between a state and the goal. For single-stage environments, we evaluate our approach using the IoU metric for a fair and consistent comparison with PlasticineLab.

## 4.2 EVALUATION OF CPDEFORM ON MULTI-STAGE TASKS

We compare our approach with the following baselines: **PlasticineLab**: The vanilla gradient-based solver does not come with any contact point discovery features. It corresponds with the stand-alone differentiable physics solver described in Huang et al. (2021). **Reinforcement Learning (RL)**: We evaluate the performance of the existing RL algorithms on our tasks. We use three model-free reinforcement learning algorithms: Soft Actor-Critic (SAC) (Haarnoja et al., 2017), Policy Proximal Optimization (PPO) (Schulman et al., 2017), and TD3 (Fujimoto et al., 2018). For each stage, we optimize for 200 episodes for differentiable physics-based approaches with a learning rate 0.1. For each environment, we modestly choose a horizon of 10 or 20. We restrict the number of environment steps used for optimization under 1 million. We train each RL algorithm on each environment for 1000 episodes, with 1000 environment steps per episode, which accounts for the 1 million environment-step limit. Our reward function is of the form $\mathcal{R} = -\mathcal{R}_{shape} - \mathcal{R}_{grasp}$, where $\mathcal{R}_{shape}$ is the Wasserstein-1 distance between the source and target particles for measuring shape differences. And $\mathcal{R}_{grasp}$ encourages the manipulators to be closer to the soft bodies. For fair comparison with Pinch from PlasticineLab, we use the reward formulation described in Huang et al. (2021).

We show the quantitative results in Table 3 and the qualitative results in Figure 6. We find that our approach is capable of finishing these complex tasks, and significantly outperforms the baselines. We find that with the discovered contact points, our approach is able to iteratively build and refine the nose, tail, and wings of the **Airplane**. In **Chair**, we find that our approach guides the solver to first create the general seat, then refine the arm rest and back of the chair. In **Bottle**, our approach first pushes down the top of the plasticine cube to create the neck, before refining the sides of the bottle. For **Move++**, our approach is able to complete the transportation tasks of the three cubes by selecting the most advantageous object to transfer at each stage. In **Rope++**, our approach first moves the rope to form the general shape, before refining the ends of the rope. In **Writer++**, our approach is capable to iteratively guide the solver to print the "ICLR" letters on the plasticine cube. Comparatively, we

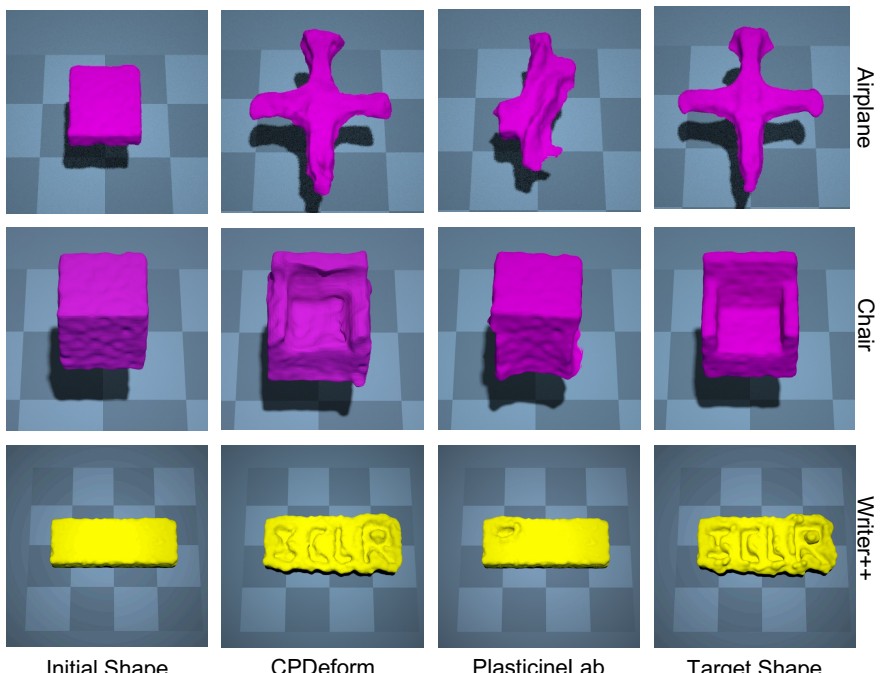

| | Initial Shape | CPDeform | PlasticineLab | Target Shape |

Figure 6: Qualitative results of CPDeform and PlasticineLab on multi-stage task environments.

| **Env** | Airplane | Chair | Bottle | Star |
|---|---|---|---|---|
| SAC | $0.0437 \pm 0.0006$ | $0.0319 \pm 0.0003$ | $0.0383 \pm 0.0020$ | $0.0382 \pm 0.0012$ |
| PPO | $0.0507 \pm 0.0082$ | $0.0329 \pm 0.0004$ | $0.0394 \pm 0.0002$ | $0.0352 \pm 0.0003$ |
| TD3 | $0.0555 \pm 0.0043$ | $0.0364 \pm 0.0033$ | $0.0443 \pm 0.0001$ | $0.0403 \pm 0.0047$ |
| PlasticineLab | $0.0324 \pm 0.0023$ | $0.0218 \pm 0.0002$ | $0.0283 \pm 0.0007$ | $0.0274 \pm 0.0003$ |
| CPDeform | $\mathbf{0.0073 \pm 0.0001}$ | $\mathbf{0.0072 \pm 0.0001}$ | $\mathbf{0.0093 \pm 0.0001}$ | $\mathbf{0.0198 \pm 0.0004}$ |
| **Env** | Move++ | Rope++ | Writer++ | Pinch |
| SAC | $0.1660 \pm 0.0185$ | $0.0333 \pm 0.0038$ | $0.0149 \pm 0.0002$ | $0.0204 \pm 0.0002$ |
| PPO | $0.1936 \pm 0.0508$ | $0.0387 \pm 0.0082$ | $0.0164 \pm 0.0001$ | $0.0189 \pm 0.0002$ |
| TD3 | $0.2164 \pm 0.0057$ | $0.0396 \pm 0.0023$ | $0.0153 \pm 0.0083$ | $0.0202 \pm 0.0003$ |
| PlasticineLab | $0.1895 \pm 0.0001$ | $0.0075 \pm 0.0004$ | $0.0122 \pm 0.0001$ | $0.0094 \pm 0.0016$ |
| CPDeform | $\mathbf{0.0127 \pm 0.0030}$ | $\mathbf{0.0052 \pm 0.0004}$ | $\mathbf{0.0067 \pm 0.0001}$ | $\mathbf{0.0081 \pm 0.0001}$ |

Table 1: The averaged Wasserstein-1 distance and the standard deviations of each method.

find that the stand-alone differentiable physics solver fails completely on the multi-stage tasks, as it lacks the necessary exploration power to overcome the local minima. Take **Move++** as an example, the vanilla solver is unable to move away from the cube it first initiates contact with after a stage. Similar to PlasticineLab (Huang et al., 2021), we also observe that the RL approaches in general perform worse than the vanilla gradient-based method.

### 4.3   EVALUATION OF CONTACT POINT DISCOVERY ON SINGLE STAGE TASKS

To further demonstrate the effectiveness of our approach, we compare the one-shot contact points discovered by the backbone of CPDeform with the human-defined contact points on single-stage tasks from PlasticineLab. Table 2 lists the normalized incremental IoU scores, together with the standard deviations of all approaches.

From Table 2 we can see that on most tasks, CPDeform performs better than or on par with the manipulation performance obtained from the human-defined initial end-effector positions from PlasticineLab.

In **Table**, the agent needs to push one of the four legs of the plasticine. The initial position of the agent in PlasticineLab does not directly establish contact between the end effectors and the plasticine. The stand-alone differentiable physics solver from PlasticineLab with contact loss is inadequate for guiding the agent to find the correct "leg" to push. Whereas in CPDeform, the optimal priorities is

| Env | Move | Tri. Move | Torus | Rope |
|---|---|---|---|---|
| PlasticineLab | **0.90 ± 0.12** | 0.35 ± 0.20 | 0.77 ± 0.39 | 0.59 ± 0.13 |
| CPDeform | 0.82 ± 0.12 | **0.63 ± 0.15** | **0.99 ± 0.01** | **0.73 ± 0.12** |

| Env | RollingPin | Chopsticks | Assembly | Table |
|---|---|---|---|---|
| PlasticineLab | **0.93 ± 0.04** | **0.88 ± 0.08** | **0.90 ± 0.10** | 0.01 ± 0.01 |
| CPDeform | 0.89 ± 0.06 | 0.34 ± 0.19 | 0.82 ± 0.13 | **0.74 ± 0.23** |

Table 2: We compare our contact point discovery backbone with human-defined contact points from PlasticineLab on single-stage tasks. We show the averaged normalized incremental IoU scores and the standard deviations of each method.

capable of identifying the "leg" of interest, resulting in a suitable placement of the manipulator that significantly improves task completion. However, we recognize that CPDeform is not a panacea. We notice that CPDeform struggles on **Chopsticks** due to the limitation of transport priorities for capturing the dramatic topological change. Specifically, we find that the transport priorities cannot discover the transport plan that preserves the continuity of the topology, as discussed in Feydy (2020). How to find a continuous correspondence to map a source shape to target remains to be an interesting future direction to explore.

### 4.4 COMPARISON WITH OTHER CONTACT DISCOVERY METHOD

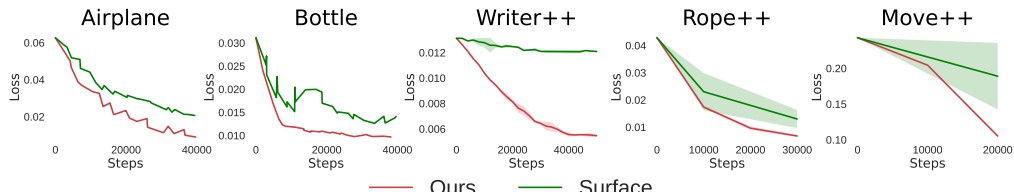

Figure 7: Ablation Learning Curves

In this section, we perform an ablation study to verify if transport priorities help find accurate solutions efficiently. We compare CPDeform with random sampling contact points on the surface of the soft bodies, while maintaining all other settings and hyperparameters, including the number of stages and candidate pose sets. Figure 7 shows the comparison of randomly sampled contact points with CPDeform. We observe that our method in CPDeform outperforms the surface-point sampling strategy by a large margin in terms of both accuracy and efficiency, proving the effectiveness of the transport-priority-based contact point discovery backbone employed by CPDeform.

## 5 RELATED WORK

**Soft Body Manipulation.** Soft body manipulation has a long history in robotics across multiple fields, such as fabric manipulation (Liang et al., 2019; Wu et al., 2020; Ha & Song, 2021), rope controlling (Yan et al., 2020; Wu et al., 2020; Gan et al., 2021), and food preparation (Bollini et al., 2013; Heiden et al., 2019). In this work, we study the elastoplastic materials in PlasticineLab (Huang et al., 2021) due to their wider applications than the pure elastic material model, leading to more general and realistic modeling of real-world soft bodies. The high degrees of freedom of soft bodies (Hu et al., 2018; Essahbi et al., 2012) limit the application of motion planning methods (Kuffner & LaValle, 2000; Kavraki et al., 1996) and most works focus on linear models like rope (Saha & Isto, 2006; 2007; Wakamatsu et al., 2006) or planar models like cloth (McConachie et al., 2020). Control methods such as Hirai & Wada (2000); Wada et al. (2001); Smolen & Patriciu (2009) bypass the expensive global planning by approximating local models and manipulating soft bodies with local controllers, which sacrifices the ability to plan for the multi-stage tasks that we study in this work.

Recent development of Deep RL (Mnih et al., 2013; Schulman et al., 2017; Haarnoja et al., 2017; Fujimoto et al., 2018) methods has enabled a unified approach to learn both perception module and manipulation policy in an end-to-end way (Lin et al., 2020; Wu et al., 2020; Nair et al., 2017; Wang et al., 2019). However, policy learning suffers from exploration issue and usually require a huge

amount of data or additional expert demonstrations to support imitation learning (Wu et al., 2020; Lee et al., 2015; Matas et al., 2018; Seita et al., 2019). Another popular approach is to use neural networks to approximate soft body dynamics (Li et al., 2018; Lin et al., 2021; Hoque et al., 2021) and solve control problems with model predictive control (Rubinstein, 1999) or gradient-based optimization. While these methods gain the generalizability by employing a shared local dynamic model, they need a way to use the model for planning. Similar to our iterative contact-and-deform approach, methods like Lin et al. (2020); Seita et al. (2019); Li et al. (2018) apply a pick-and-place action space to manipulate soft bodies like cloth and rope. While this action space enables efficient exploration and simplifies learning tasks, it poses a restriction for the type of end effectors, making it unsuitable for the fine-grained manipulation of soft bodies such as plasticine. Comparatively, CPDeform leverages the differentiable physics solver to complete more sophisticated manipulation tasks.

**Differentiable Physics for Trajectory Optimization.** Our method uses the differentiable simulator developed in PlasticineLab (Huang et al., 2021) for trajectory optimization. Encouraged by the success of gradient descent in neural network learning, differentiable physics with analytic physics models (Geilinger et al., 2020; Degrave et al., 2016; de Avila Belbute-Peres et al., 2018; Carpentier & Mansard, 2018; Giftthaler et al., 2017; Heiden et al., 2019; 2020; 2021; Toussaint et al., 2018; Hu et al., 2019; 2020; Qiao et al., 2020; Murthy et al., 2020; Millard et al., 2020; Werling et al., 2021; Du et al., 2021; Ma et al., 2021) has gained increasing popularity. As mentioned in Sec. 2, applying gradient-optimization with an inappropriate contact point would get stuck in the local minima, especially in tasks where multiple contact switches are needed. Previous research has explored various ways to handle contacts for rigid bodies. Contact-invariant optimization (Mordatch et al., 2012) imposes variables as soft representation of objects' contact relationship, and in (Posa et al., 2014; Sleiman et al., 2019), contacts are handled implicitly within analytical models. However, due to the nonlinear and contact-rich nature of manipulation tasks, we often have to combine search and optimization to solve mixed-integer program (Han & Tedrake, 2020) or logic-geometric program (Toussaint et al., 2018) problems, which are inefficient given the complexity of 3D soft bodies. We explore a complementary direction of previous approaches. By integrating visual cues into differentiable physics, we are able to skip many local minima and boost the performance in various soft body manipulation tasks.

**Grasping and Rigid Body Manipulation.** Our contact point discovery approach shares a similar spirit with grasp pose detection in rigid body manipulation. Determining contact point or grasp pose for different end effectors is one of the everlasting topics in rigid body manipulation (Bohg et al., 2014; Miller & Allen, 2004; Dang & Allen, 2012; Qin et al., 2020; Mahler et al., 2017). By analyzing 3D geometry (Sahbani et al., 2012; Hong et al., 2021), one can find antipodal grasps that satisfy force closure (Nguyen, 1988; Chen & Burdick, 1993), and grasp objects without running simulation. Similarly, our contact point discovery method tries to find contact points through geometrical analysis, in an effort to extend geometric analysis for more general soft body manipulation tasks.

## 6 Conclusions and Future Work

In this paper, we propose a novel framework, CPDeform, that integrates optimal transport-based contact discovery into differentiable physics. Extensive experiments suggest that our proposed contact point discovery method, when directly employed by the differentiable solver, performs on par with or better than human-defined initial contact points on single-stage tasks. On multi-stage tasks that are infeasible for the vanilla solver, CPDeform employs a heuristic searching approach to iteratively solve the tasks. Our work demonstrates the importance of contact points in policy learning with differentiable physics and the advantage of geometric-analysis methods as a heuristic.

Our framework requires a properly defined Wasserstein distance on the object's representation. The choice of material types does not affect our optimal transport heuristic because it relies on shape information only. We assume uniform density of the object template and moderately similar topology across initial and target shapes. Interesting avenues for future work include generalizing the discovery of useful contact points through learning methods for a diverse set of shapes, and applying a similar contact point discovery principle to dexterous rigid body manipulation, or combining it with other planning approaches. We refer the readers to Appendix G for more discussions on the limitation and future works.

**Acknowledgement.** We thank Hannah Skye Dunnigan for her help on graphic design. This work was supported by MIT-IBM Watson AI Lab and its member company Nexplore, ONR MURI (N00014-13-1-0333), DARPA Machine Common Sense program, ONR (N00014-18-1-2847) and MERL.

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

# A    ENVIRONMENT DETAILS

- **Airplane** The agent needs to manipulate a pair of "spatulas" (represented as round boxes) to sculpt the nose, wing, and tail on the plasticine cube to match the target airplane shape.
- **Chair** With a pair of "spatulas," the agent needs to build the seat, back, and armrest from the plasticine cube to reach the target chair shape.
- **Bottle** With a pair of "spatulas," the agent needs to build the neck and body from the plasticine cube to reach the target bottle shape.
- **Star** The agent needs to manipulate a pair of "spatulas" to build the tips of the star from the plasticine cube to reach the target bottle shape.
- **Move++** The agent needs to use a single pair of sphere manipulators to separately transport three plasticine cubes to fulfill the three destinations of demand.
- **Rope++** The agent needs to use a single pair of sphere manipulators to reshape the plasticine rope to reach the target polyline shape.
- **Writer++** The agent manipulates a "pen" (represented using a vertical capsule) to print the letters "ICLR" on the plasticine cube.
- **Pinch** In this task, the agent manipulates one rigid sphere to create dents specified by the target shape on the plasticine box.

# B    MORE DETAILS ABOUT OPTIMAL TRANSPORT

Optimal transport is useful for comparing measures $\mu$ and $\nu$ in a Lagrangian framework, by accounting for the cost of transporting one measure to another. Here, we consider the first measure $\mu$ as the soft-body particles in the source state and $\nu$ as the soft-body particles in the target state that we wish to achieve through manipulation, where both input measures are on the space $\mathbb{R}^{N_p \times 3}$. Let $\alpha, \beta \in \mathbb{R}^{N_p}$ be the mass vectors that for the particles in $\mu, \nu$ respectively. Since all particles are weighted equally in our simulator, $\alpha = \beta = \mathbf{1}$. Let $U(\alpha, \beta)$ be the polytope that contains all transport plans, which are non-negative matrices whose rows and columns sum to $\alpha, \beta$ respectively. We have $U(\alpha, \beta) := \{P \in \mathbb{R}_+^{N_p \times N_p} \mid P\mathbf{1} = \alpha, P^T\mathbf{1} = \beta\}$. Given a cost function $C : \mathbb{R}^3 \times \mathbb{R}^3 \mapsto [0, \infty]$, we create the cost matrix $M \in \mathbb{R}^{N_p \times N_p}$, where each element $M_{i,j} = C(\mu_i, \nu_j)$ measures the cost of transporting the particle $\mu_i$ to $\nu_j$. Now, the cost of mapping $\mu$ to $\nu$ can be quantified as $\langle P, M \rangle$. We define the primal form of the *optimal transport* (OT) problem as $\text{OT}(\alpha, \beta) := \min_{P \in U(\alpha, \beta)} \langle P, M \rangle$.

# C    PSEUDOCODE OF CPDEFORM

---
**Algorithm 1** CPDeform
---
**Input:** Current shape $S_0$ with $N$ particles, goal shape $G$, number of stages $n_{stage}$, number of steps per stage $n_{step}$, candidate pose set $\mathcal{T}$
1: **for** $0 \leq i < n_{stage}$ **do**
2:     Compute optimal transport priorities $\{\alpha_i\}_{i \leq N}$ between $S_i$ and goal shape $G$
3:     Find point $p$ with largest priorities $\alpha_p$
4:     **for** pose $t \in \mathcal{T}$ **do**
5:         Place manipulators around point $p$ with largest heuristic value without collisions
6:         Solve trajectories with differentiable physics to generate plan $p_t$ and compute its shape matching loss $c_t$
7:     **end for**
8:     Execute the plan $p_t$ with the minimal loss $c_t$
9: **end for**
---

# D    CONTROLLER OPTIMIZATION WITH DIFFERENTIABLE PHYSICS

Consider that the motion trajectory is parameterized as a sequence of three-dimensional actions $\{a_1, \ldots, a_T\}$, where $T$ is the number of simulation steps. Let $\{s_t\}_{0 \leq t \leq T}$ be simulation states at

different time steps, which include the state of the plasticine and manipulator. The differentiable simulator starts from the initial state $s_0$ and completes the action sequences by repeatedly executing the forward function $s_{t+1} = \phi(s_t, a_{t+1})$ until the ending state $s_T$ has been reached. The objective of the optimizer is to minimize the distance between the final shape and the target shape. We represent the objective as a loss function $\mathcal{L}(s_T, g)$ where $g$ is the target shape. Since the simulation forward function $\phi$ is fully-differentiable, we can compute gradients $\partial\mathcal{L}/\partial a_t$ of the loss $\mathcal{L}$ with respect to each action $a_t$, and run gradient descent steps to optimize the action sequences by $a_t = a_t - \alpha\partial\mathcal{L}/\partial a_t$ where $\alpha$ is the learning rate. We repeat this optimization loop for $N$ iterations. The overall process is reflected by Algorithm 2.

---

**Algorithm 2** Differentiable Physics Solver for Controller Optimization

---

**Input:** Target shape $g$, action sequence $\{a_1, \ldots, a_T\}$, initial state $s_0$, differentiable forward simulation function $\phi$, number of iterations $N$, learning rate $\alpha$
1: **for** $0 \leq i < N$ **do**
2:     **for** $0 \leq t < T$ **do**
3:         $s_{t+1} = \phi(s_t, a_{t+1})$
4:     **end for**
5:     Compute shape difference loss using $\mathcal{L}(s_T, g)$
6:     Update the action sequence by $a_t = a_t - \alpha\partial\mathcal{L}/\partial a_t$
7: **end for**

---

# E   ADDITIONAL BASELINES

We compare CPDeform with two additional baselines, named Multiple-Restarts (Multi-Re) and Bayesian-Optimization (Bayes-Op).

**Investigation on multiple restarts** For each stage, we randomly sample a collection of 15 contact plans. We then use the differentiable physic solver to optimize the action sequence. The execution of the solver for each plan corresponds with a single restart or an "initial guess." We use the contact plan that achieves the lowest loss for manipulation.

**Investigation on Bayesian optimization** For each stage, we use Bayesian optimization with Gaussian process to optimize the contact plan. We employ a black box function that takes a contact plan as its input and outputs the loss achieved by the differentiable physics solver with that contact plan. We then perform Bayesian optimization for 15 iterations to optimize the contact plan for the given stage.

| **Env** | Airplane | Chair | Bottle | Star | Move++ | Rope++ | Writer++ | Pinch |
|---|---|---|---|---|---|---|---|---|
| Multi-Re | 0.0122 | 0.0098 | 0.0112 | 0.0221 | 0.1209 | 0.0055 | 0.0076 | 0.0094 |
| Bayes-Op | 0.0369 | 0.0103 | 0.0136 | 0.0212 | 0.1234 | 0.0067 | 0.0113 | 0.0092 |
| PlasticineLab | 0.0324 | 0.0218 | 0.0283 | 0.0274 | 0.1895 | 0.0075 | 0.0122 | 0.0094 |
| CPDeform | **0.0073** | **0.0072** | **0.0093** | **0.0198** | **0.0127** | **0.0052** | **0.0067** | **0.0081** |

Table 3: The averaged Wasserstein-1 distance of each method.

In general, we observe that Multiple-Restarts and Bayesian-Optimization perform better than PlasticineLab, or the standalone differentiable physics solver. CPDeform outperforms these two additional baselines. Additionally, because both Multiple-Restarts and Bayesian-Optimization rely heavily on trials and errors to find the suitable contact points, as illustrated in Figure 8, they are much more computationally expensive and less efficient than CPDeform.

# F   RUNTIME

We show the runtime of CPDeform, PlasticineLab, Bayesian-Optimization, and Multiple-Restarts in Figure 8. RL approaches are not drawn because they cannot complete the manipulation tasks, and their loss curves are close to the starting loss. We draw the wall-time in seconds on the x-axis and the corresponding Wasserstein-1 distance loss values on the y-axis. Each data point represents the lowest

loss achieved for that method at the end of a stage. Bayesian-Optimization and Multiple-Restarts spend more time than the CPDeform and PlasticineLab on each stage due to the number of Bayesian optimization iterations and the number of restarts performed, respectively. We observe that CPDeform is more efficient than other approaches.

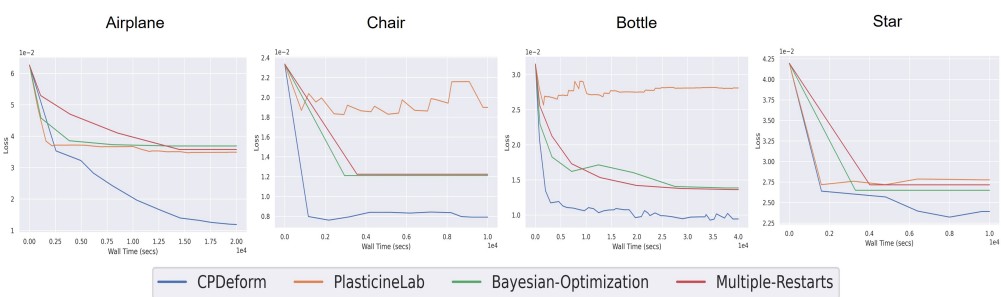

Figure 8: Runtime of each method. We draw the wall-time in seconds on the x-axis and the corresponding Wasserstein-1 distance loss values on the y-axis. Each data point represents the lowest loss achieved for that method at the end of a stage.

## G  LIMITATION AND FUTURE WORKS

In this section, we discuss several implicit assumptions we made when constructing tasks. These limitations are not common in our tasks and could be addressed by including additional techniques; we did not take those factors into account in this work. We believe they constitute several interesting future directions to explore.

**Density Assumption** Low or non-uniform density could potentially impact the optimization steps due to varying optimal transport mapping. For instance, the optimal transport loss might not be able to capture areas of the target shape whose particle distributions are extremely sparse (e.g., a few particles) and is likely unable to guide the differentiable physics solver to task completion. In such a case, having low or non-uniform density might pose a limit on our method. As a solution, we have uniformly sampled particles inside the target shapes. By uniformly sampling particles inside a given 3D shape of mesh representation, we create its corresponding target template that the differentiable physics solver can use for the manipulation tasks. We think that considerations of target shapes (e.g., a cuboid) with varying internal configurations (e.g., non-uniform density) is an interesting future direction.

**Volume Assumption** We assume that the volumes of the initial and target shapes are similar. Since our tasks use manipulators that are tailored to compressing instead of expanding the soft materials, the initial shape cannot be significantly smaller than the target shape. Conversely, if the initial shape is significantly larger than the target shape, it will require a lot of effort to compress the initial shape into the target shape. As a solution, we can meet this condition by cutting and assembling the initial plasticines.

**Topology Assumption** We assume that the initial and target shapes do not have tremendous topological differences while requiring shape transformation that involves complex motion planning. When this condition is false, the greedy algorithm that finds the best contact points at the current stage could be limited. For example, consider a task where the initial shape is a straight rope, and the target shape is a rope with knots. In order to complete the task, the rope needs to be folded and wrapped, which might lead to a temporary increment of shape-matching loss for a few stages. The greedy approach is limited because it might be unwilling to accept the temporary increment of the loss. It would potentially arrive at a solution whose general shape is matched with the target rope by omitting the detailed knots. As a possible solution, if we incorporate RL-based algorithms into our approach, the ability to plan globally could allow the differentiable physics solver to overcome the temporary increment of the loss at an intermediate stage and act for the best completion of the task at the final stage.

# H INVESTIGATION ON THE OPTIMALITY OF CPDEFORM

We use **Writer** to study the optimality of our optimal-transport-based heuristic. Holding the height of the manipulator at a fixed value, as illustrated in the leftmost figure in Figure 9, we create a 2D grid that covers the top surface of the plasticine cube. The grid is of size 20×20, and its points are equidistant from each other on each row or column. We then record the loss achieved by executing the differentiable physics solver at each grid point (i.e., contact point). Having collected a grid of loss values, we use interpolation to draw the 2D loss landscape, as illustrated in Figure 9. By exhaustively searching through each grid point, we observe that the higher transport priorities indicated by our heuristic tend to correlate with lower loss values, and vice versa.

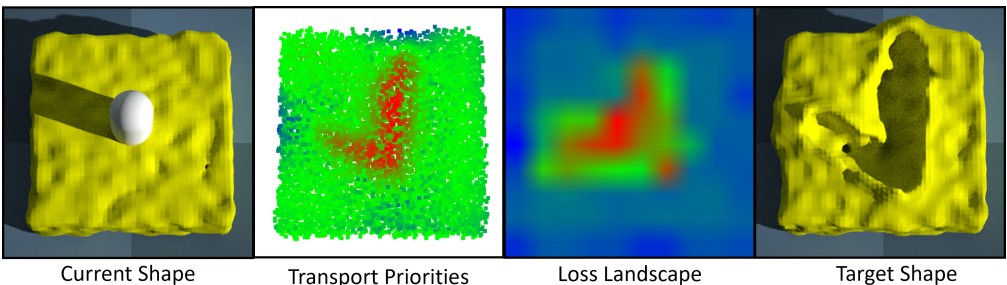

Current Shape     Transport Priorities     Loss Landscape     Target Shape

Figure 9: We demonstrate the visualizations of the transport priorities and the loss landscape using **Writer**. For the loss landscape, the color of each pixel corresponds with the loss achieved by applying the differentiable physics solver near the corresponding simulator coordinate, where the height of the manipulator is set as a fixed value (illustrated in the current shape). Here, warmer colors indicate smaller loss values and colder colors indicate larger loss values.

