# OpenReview forum: "Contact Points Discovery for Soft-Body Manipulations with Differentiable Physics"
_ICLR.cc/2022/Conference — ICLR 2022 Spotlight_

### Official Review · Reviewer_P7JN · 2021-11-01

**Correctness:** 3
**Technical Novelty And Significance:** 3
**Empirical Novelty And Significance:** 3
**Recommendation:** 8
**Confidence:** 3

**Main Review:**

The paper is in general well written and the proposed method produces quite interesting results. However, I do have some questions that I hope the authors could help address:

1) In the single manipulator placement algorithm: is the d() function the distance between the particle to the closest point on the manipulator, or the center of the manipulator? Also, is orientation considered during the placement optimization?

2) How does the algorithm determine when to switch to the next stage during a multi-stage task? Is it time-based, state-based, or reward-based?

3) For the task where the deformable object is translated/transported to a different location, does all particles receive the the same priority score? If so, how does the algorithm figure out that the best contact points? For example, if the deformable object is very flexible and the best contact point is to hold it from bottom. More generally, it’s not clear whether the current approach can take physical information into account.

4) The heuristics approach of contact selection seems reasonable in general. It would be helpful if some ground-truth results could be shown to further support the optimality of the result, e.g. by comparing to an exhaustive grid search approach.

5) The paper currently did not talk about how the controller optimization with differentiable physics is done. Although it’s mostly prior work, it would be nice to include some descriptions, maybe in appendix, to make the paper more self-contained.


**Summary Of The Paper:**

The paper proposed an algorithm to discover appropriate contact points for deformable object manipulation. A key component of the proposed algorithm is to use an optimal-transport approach that computes a transport priority score for each particle in the deformable body. This score is then used to guide a grid search procedure to determine the best initial contact point for the manipulation. For multiple manipulators, a pre-defined set of poses are enumerated to find the best pose to use. The proposed algorithm is evaluated on tasks that requires a single manipulation motion or a sequence of motions to shape a deformable object into desired shapes. The result shows improved performance compared to prior methods and enables completion of novel tasks that prior methods failed.

**Summary Of The Review:**

I think the paper presents interesting work that achieves non-trivial results in deformable object manipulation. Some of the details are missing from the text as discussed above, which would help better gauge the impact of the work and help reproduce the work by other researchers. Overall, I think the work makes a good contribution to the conference.

---

> ### Author Response · Authors · 2021-11-18
> **Response to Reviewer P7JN: Revision Updated**
>
> Thanks so much for your constructive comments and suggestions.
>
>
> > #### 1. In the single manipulator placement algorithm: is the d() function the distance between the particle to the closest point on the manipulator, or the center of the manipulator?
>
> The $d(\cdot, \cdot)$ function is the distance between the particle to the closest point on the manipulator. We have clarified this in Section 3.2 of our revised paper.
>
> > #### 2. Is orientation considered during the placement optimization?
>
> We do consider orientation during the placement optimization. This is achieved by considering a candidate pose set, where each pose corresponds with a distinct orientation (Figure 4, bottom). We execute differentiable physics solver with each pose and select the one with minimal loss.
>
> > #### 3. How does the algorithm determine when to switch to the next stage during a multi-stage task? Is it time-based, state-based, or reward-based?
>
> In our method, determining when to switch to the next stage could be considered time-based. During the execution of the differentiable physics solver, we optimize for a fixed number of iterations. During the manipulation of the soft bodies in each stage, the action sequence consists of a fixed number of simulation steps. We think this fits the definition of "time-based."
>
>
> > #### 4. About the priority scores for translation and physical understanding.
>
> - In this work, **our goal is to leverage visual appearances to guide the differentiable physics solver through geometric analysis**. We agree that the inclusion of physical attributes is exciting and could further extend our approach.
> - For tasks where the deformable object is translated to a different location, particles do receive nearly the same priority scores. In this case, we do need to consider the physical information.
> - In our work, **by searching over the candidate pose set of the manipulators, we have implicitly taken the physical information into account**. This is because different candidate poses correspond with different manipulation strategies and contact points.
>   - For example, in the multiple-manipulator case, the candidate pose set covers left-right, top-bottom, and front-back grasping poses. By enumerating over these grasping poses and executing the differentiable physics solver, our approach chooses the most beneficial candidate pose and contact points with the lowest loss. During this process, we have naturally leveraged the physical dynamics through differentiable physics optimization.
>   - Our approach would thus arrive at the candidate pose that best takes advantage of the physical information. Consider the task of transporting an object straight up, our pose selection will likely choose the top-bottom grasping pose based on the physical settings of the simulated environment.
> - We think that **searching over a more diverse set of poses can further solve this problem**.
>   - Specifically, we can search over numerous poses and choose the one with the largest decrease of loss, where we have considered the underlying physical dynamics through backpropagation supported by the differentiable simulation. Additionally, to speed up this search, we can employ heuristics based on geometric analysis.
>   - However, since we find that the current candidate pose set is sufficient enough to achieve promising results and does not need to be exhaustively large in practice, we did not include this strategy into our approach.
>
>
> > #### 5. Showing ground-truth results to further support the optimality of the method.
>
> Thanks for your detailed comments! We have added Figure 9 in Appendix H to further investigate and support the optimality of the heuristic.
>
> > #### 6. Adding descriptions on the controller optimization with differentiable physics.
>
> We have added descriptions on the controller optimization with differentiable physics in Appendix D of our revised paper.
>
> > #### 7. Some of the details are missing from the text as discussed above, which would help better gauge the impact of the work and help reproduce the work by other researchers.
>
> Thank you very much for your detailed comments! We have added the suggested details in our revised paper, and we promise to release source codes for other researchers to reproduce our work.
>
>
>
>
> **We hope that our response has addressed your concerns, and would really appreciate it if you could raise your rating.** *Please do not hesitate to contact us if there are other clarifications or experiments we can offer.*
>
> Best, Authors

---

> > ### Comment · Reviewer_P7JN · 2021-12-01
> > **Thanks for the rebuttal**
> >
> > Thank you for your response. The rebuttal addressed most of my comments and thus I'd be happy to increase my recommendation.
> >
> > I'm not fully convinced about the argument for implicitly taking physics information into account though. Similar to the example provided by the authors, if the task is to translate an object horizontally, a physics-aware controller will select contact points from the bottom to hold the object, while the current geometry-only algorithm will select contact points from the side. It would be helpful to make this detail clear in the paper, which could also inspire future work directions.

---

> > > ### Author Response · Authors · 2021-12-02
> > > **Thank you**
> > >
> > > Thank you for your constructive comments and suggestions. We will follow your recommendation and clarify the detail about physical awareness in the final version of the paper. Thank you again for your time.
> > >
> > > Best, Authors

---

### Official Review · Reviewer_Ept9 · 2021-11-03

**Correctness:** 4
**Technical Novelty And Significance:** 3
**Empirical Novelty And Significance:** 3
**Recommendation:** 8
**Confidence:** 3

**Main Review:**

This method design an automatic strategy to stage long manipulation tasks and discover the contact points for each task. The use of optimal transport priority as a heuristic for contact areas is reasonable and intuitive. The same heuristic is also used in DiffAqua, the authors might want to cite and discuss their similarities.  As far as I can tell from the demo video, this method can successfully perform long manipulation tasks that I have never seen in other differentiable physics papers, which is impressive. However, the optimal transport heuristic might be only suitable for these MPM-based objects, it remains interesting how to apply this method to rigid or elastic soft materials.

Some concerns that I have,
1. What are the definitions of $\alpha, \beta$ in Sec. 3.1?
2. This is actually a greedy algorithm, finding the best contact areas at the current stage and deforming the object to the target shape as close as possible. Without global planning, is it possible that the schedule is suboptimal or can not even converge？
3. How long it will take to find an optimal motion path?



**Summary Of The Paper:**

This paper proposes a method to solve the multistage manipulation tasks for soft materials. Differentiable physics has been used in controlling and manipulating soft materials since DiffTaichi. However, optimization methods based on local gradient information can be easily trapped in local minima. This paper divides a long task into small stages, within which the contact area will remain unchanged. The contact points are selected by choosing the regions with the largest transport priorities. The end effectors are then placed with poses from the candidate set. After running trajectory optimization using physics gradients, the final pose is finalized as the pose with minimum loss.

**Summary Of The Review:**

This paper proposes an intuitive and effective way to utilize differentiable physics in long-horizon motion planning for soft materials. Compared to previous papers like DiffTaichi and PlasticineLab, this method can better handle multistage tasks. The demo videos show its capacity to plan very long tasks (up to 140s).

---

> ### Author Response · Authors · 2021-11-18
> **Response to Reviewer Ept9: Revision Updated**
>
> Thanks so much for your detailed and constructive comments.
>
> > #### 1. The same heuristic is also used in DiffAqua, the authors might want to cite and discuss their similarities.
>
>
> #### 1.1 Similarities between our work and DiffAqua.
>
> Thank you for bringing DiffAqua to our attention. We agree that DiffAqua and our work share some similarities. Indeed, both works use optimal transport as a heuristic and combine it with differentiable simulation.
>
> #### 1.2 Differences between our work and DiffAqua.
>
> The difference is that DiffAqua uses optimal transport to interpolate shapes in a library, and we use optimal transport as a heuristic to propose contact point locations. **The two papers are relevant, but their problem domains are different**. We will cite and discuss DiffAqua in our revised manuscript.
>
>
> > #### 2. The optimal transport heuristic might be only suitable for these MPM-based objects, it remains interesting how to apply this method to rigid or elastic soft materials.
>
> #### 2.1 Extension to other simulation methodologies is an interesting future direction.
>
> Indeed, we demonstrate the optimal transport heuristic using MPM simulation of elastoplastic materials only, and we agree that extending our method to other simulation methodologies (e.g., mesh-based, voxel-based, or level-set-based simulation) different from MPM and other material types (rigid or elastic as you suggested) is an interesting future direction.
>
> #### 2.2 MPM-based object representation is not required.
>
> However, we would like to mention that our optimal transport heuristic does not require an object to be represented and simulated by MPM. **Instead, it requires a properly defined Wasserstein distance on the object's representation, regardless of whether it is material points in MPM, voxels, or meshes**. One example is the DiffAqua paper mentioned above, which uses an optimal transport heuristic on voxel-based objects. We have clarified this assumption of our optimal transport heuristic in Section 6 of the manuscript.
>
> #### 2.3 Optimal transport heuristic is not affected by the choice of material types.
>
> We would also like to mention that **the choice of material types (rigid, elastic, or elastoplastic) does not affect our optimal transport heuristic because it relies on shape information only**. Instead, the differentiable simulator in our method depends on these material types. Fortunately, quite a few differentiable rigid-body and elastic-body simulators exist, and it would be interesting to replace the elastoplastic differentiable simulator in our pipeline with them. We have added a brief discussion on this topic in Section 6 of our revised paper.
>
>
> > #### 3. Definitions of $\alpha, \beta$ in Sec. 3.1.
>
> We thank the reviewer for pointing out the missing definition. $\alpha, \beta$ are the mass vectors that for the particles in $\mu, \nu$ respectively. We have added this back from Appendix B to clarify the details in Section 3.1 of our revised paper.
>
> > #### 4. About the suboptimality of our approach.
>
> While we realize that the greedy approach is suboptimal, **the focus of our work is on the effectiveness of contact point discovery for soft-body manipulation tasks**. We find that the greedy algorithm is sufficient enough to achieve promising results when combined with contact point discovery.
>
> #### An example of suboptimality and its potential solution.
>
> - Consider a task where the initial shape is a straight rope, and the target shape is a rope with knots. In order to complete the task, the rope needs to be folded and wrapped, which might lead to a temporary increment of shape-matching loss for a few stages. The greedy approach is suboptimal because it might be unwilling to accept the temporary increment of the loss. It would potentially arrive at a solution whose general shape is matched with the target rope by omitting the detailed knots.
> - As a possible solution, **global planning can be integrated into our framework by adding RL-based algorithms into our approach**, which is an interesting future direction.
>
> > #### 5. About the convergence of our approach
>
> The convergence depends on the differentiable physics solver we use. Our greedy schedule ensures the loss is non-increasing if the differentiable physics solver can find a solution that does not increase the shape differences and thus converges. In practice, we notice that the differentiable physics solver converges well, and our approach eventually converges around at local minima.
>
>
> > #### 6. Runtime of the approach.
>
> We have added the runtime plots in Figure 8 in Appendix F.
>
> *We sincerely appreciate your comments. Please feel free to let us know if you have further questions.*

---

### Official Review · Reviewer_rUod · 2021-11-08

**Correctness:** 3
**Technical Novelty And Significance:** 2
**Empirical Novelty And Significance:** 3
**Recommendation:** 6
**Confidence:** 4

**Main Review:**

### Strengths

* **S1** This paper addresses an important gap in differentiable physics engines. Existing differentiable physics engines (Degrave et al., Qiao et al., and others) provide gradients through the physical simulation steps (i.e., evolution of state variables like velocity/acceleration/displacement) with respect to chosen initial physical parameters (forces/torques etc.). However, they provide accurate gradients for a task only if the contacts required to accomplish the task already are known a priori. To the best of my knowledge, there isn’t an approach that also differentiates through the ‘contact generation’ step, which involves reasoning over the space of all possible contacts.
* **S2** This paper takes an optimal transport based approach, which is theoretically (physically) motivated and has advantages of simplicity.
* **S3** The proposed approach significantly outperforms a gradient descent baseline (differentiable simulation without differentiating through contact generation) and select reinforcement learning baselines.
* **S4** The paper is well-written and easy to follow


### Weaknesses

* **W1** This paper seems to make a few implicit assumptions that potentially have ramifications on whether this will generalize to noisy or simply incorrect object templates. I believe this requires some discussions, likely in a limitations (sub-)section. For instance, if the target trajectory is obtained from an object with similar external appearance (e.g., a cuboid) but with very different internal configuration (low or non-uniform density opposed to a uniform density structure), one would expect optimal transport mappings to vary significantly. This may in-turn impact subsequent optimization steps
* **W2** In terms of baselines, the current manuscript compares with a gradient-based optimization method (PlasticineLab) and three RL approaches (SAC, PPO, TD3). However, as the primary aim of this paper is to design a gradient-based solution to multi-step contact-rich manipulation tasks, I believe there are other gradient-based variants that can be evaluated in this setup. One immediate approach that comes to mind is gradient descent with multiple restarts (i.e., differentiable simulation based optimization with multiple initial guesses opposed to a single guess). Another approach might also be to leverage Bayesian optimization for this task.
* **W3** From the paper, it is unclear as to how the runtime of the approaches vary — reporting runtime overhead for each baseline would help better contextualize this paper


### Minor remarks

These are nitpicks; I believe these are easily addressed in a minor revision. The authors need not respond to these in the rebuttal phase.

* At multiple places in the manuscript, “differential” has been the word used to describe what I believe should be “differentiable” (e.g., differential simulator -> differentiable simulator)
* Penultimate paragraph of page 3: “Such phenomena are commonly observed across soft body manipulation tasks using the differentiable physics solver.” Perhaps reference a concrete published work to corroborate this claim?
* The best-performing approaches may be highlighted (boldface-d) in table 2



**Summary Of The Paper:**

This paper presents an optimal transport based solution to differentiate through the process of contact generation in differentiable physics engines. This enables solving contact-rich multi-step manipulation tasks for deformable bodies.


**Summary Of The Review:**

Overall, I believe this paper tackles an important and interesting problem in the differentiable physics community. While I believe the proposed approach has merit, I would like to see some of the aforementioned issues (particularly discussion of failure modes and choice of baselines) before I can recommend acceptance.

---

> ### Author Response · Authors · 2021-11-18
> **Response to Reviewer rUod: Revision Updated**
>
> Thanks so much for your interest in our method and your constructive comments.
>
>
> > #### 1. This paper seems to make a few implicit assumptions that potentially have ramifications on whether this will generalize to noisy or simply incorrect object templates. I believe this requires some discussions, likely in a limitations (sub-)section.
>
> We thank the reviewer for raising this important issue.
>
> #### 1.1 leveraging external appearances vs. internal configurations.
> - We would like to highlight that our goal is to leverage the external visual appearances instead of the internal configurations of the object templates.
> - Our tasks do not require us to know the internal structures of the object templates. **Thus, in this work, we assume that object templates have uniform densities**. By uniformly sampling particles inside a given 3D shape of mesh representation (e.g., chair), we create its corresponding object template that the differentiable physics solver can use for the manipulation tasks.
> - We agree that considerations of object templates (e.g., a cuboid) with varying internal configurations (e.g., non-uniform density) could lead to a very interesting manipulation problem, which is indeed an important future direction.
>
>
> #### 1.2 Potential impact and limitation of object templates with low or non-uniform density.
>
> - Low or non-uniform density could potentially impact the optimization steps due to varying optimal transport mapping.
> - For instance, the optimal transport loss might not be able to capture areas of the target shape whose particle distributions are extremely sparse (e.g., a few particles) and is likely unable to guide the differentiable physics solver to task completion. In such a case, having low or non-uniform density might pose a limit on our method.
> - As a solution, we have uniformly sampled particles when creating object templates.
>
>
> #### 1.3 Identification of implicit assumptions and potential solutions.
>
> - First, we assume that **the volumes of the initial and target shapes are similar**.
>     - Since our tasks use manipulators that are tailored to compressing instead of expanding the soft materials, the initial shape cannot be significantly smaller than the target shape.
>     - Conversely, if the initial shape is significantly larger than the target shape, it will require a lot of effort to compress the initial shape into the target shape.
>     - As a solution, **we can meet this condition by cutting and assembling the initial plasticines**.
> - Second, we assume that **the initial and target shapes do not have tremendous topological differences** while requiring shape transformation that involves complex motion planning.
>   - When this condition is false, the greedy algorithm that finds the best contact points at the current stage could be  limited.
>   - For example, consider a task where the initial shape is a straight rope, and the target shape is a rope with knots.
>     - In order to complete the task, the rope needs to be folded and wrapped, which might lead to a temporary increment of shape-matching loss for a few stages.
>     - The greedy approach is limited because it might be unwilling to accept the temporary increment of the loss. It would potentially arrive at a solution whose general shape is matched with the target rope by omitting the detailed knots.
>   - As a possible solution, **if we incorporate RL-based algorithms into our approach, the ability to plan globally could allow the differentiable physics solver to overcome the temporary increment of the loss at an intermediate stage** and act for the best completion of the task at the final stage.
> - **Since these limitation cases are not common in our tasks**, we did not take those factors into account in this work. **We believe they constitute an interesting future direction to explore**. We have included the discussion of these implicit assumptions and their potential limitations in Appendix G.
>
> (to be continued)

---

> > ### Author Response · Authors · 2021-11-18
> > **Response to Reviewer rUod: Revision Updated (continued)**
> >
> > > #### 2. Comparison with other gradient-based variants.
> >
> > Thanks for suggesting adding other baselines. We think that these baselines can indeed further strengthen our work. We have conducted experiments for both baselines and have incorporated results into our revised paper. The experimental results are also included below.
> >
> > - Investigation on multiple restarts
> >   - For each stage, we randomly sample a collection of 15 contact plans.
> >   - We then use differentiable physic solver to optimize the action sequence.
> >   - The execution of the solver for each plan corresponds with a single restart or an "initial guess."
> >   - We use the contact plan that achieves the lowest loss for manipulation.
> > - Investigation on Bayesian optimization
> >   - For each stage, we use Bayesian optimization with Gaussian process to optimize the contact plan.
> >   - We employ a black box function that takes a contact plan as its input and outputs the loss achieved by the differentiable physics solver.
> >   - We then perform Bayesian optimization for 15 iterations to optimize the contact plan for the given stage.
> >
> >
> > We report the averaged Wasserstein-1 distance of each method. This is reflected by Table 3 in Appendix E.
> >
> > | Env                   | Airplane      | Chair       | Bottle     |  Star           |
> > | ----------------------| ------------- | --------    | ------     | --------------- |
> > | Multiple-Restarts     |  0.0122       |    0.0098   | 0.0112     |   0.02216       |
> > | Bayesian Optimization |  0.0369       |    0.0103   | 0.0136     |   0.02115       |
> > | PlasticineLab         |  0.0324       |    0.0218   | 0.0283     |   0.0274        |
> > | CPDeform              |  **0.0073**   |  **0.0072** | **0.0093** |   **0.0198**    |
> >
> >
> > | Env                   | Move++      | Rope++      | Writer++   | Pinch       |
> > | ----------------------|  --------   | --------    | --------   | ------      |
> > | Multiple-Restarts     |   0.1209    |  0.0055     | 0.0076     |  0.0094     |
> > | Bayesian Optimization |  0.1234     |  0.0067     | 0.0113     |  0.0092     |
> > | PlasticineLab         |   0.1895    |  0.0075     | 0.0122     |  0.0094     |
> > | CPDeform (ours)       |  **0.0127** |  **0.0052** | **0.0067** |  **0.0081** |
> >
> > In general, we observe that **Multiple-Restarts and Bayesian-Optimization perform better than PlasticineLab** and that **CPDeform outperforms these two new baselines**. Additionally, **because both Multiple-Restarts and Bayesian-Optimization rely heavily on trials and errors to find the suitable contact points**, as illustrated in Figure 8 in Appendix F, **they are much more computationally expensive and less efficient than CPDeform**.
> >
> >
> > > #### 3. Runtime of the approaches.
> >
> > Thanks for the suggestion. We have added the runtime plots in Figure 8 in Appendix F.
> >
> > > #### 4. Typos for "differentiable," adding a reference for the penultimate paragraph of page 3, and highlighting the best-performing approaches in table 2.
> >
> > Thank you for the detailed comments! We have fixed typos you pointed out and  addressed the suggestions in our revised paper.
> >
> >
> > **We hope that our response has addressed your concerns, and turns your assessment to the positive side.**  *Please do not hesitate to contact us if there are other clarifications or experiments we can offer.*
> >
> >
> >
> > Best, Authors

---

> > > ### Comment · Reviewer_rUod · 2021-11-22
> > > **Additional discussion, baselines add value**
> > >
> > > Dear authors,
> > >
> > > Thanks for your response. My primary concerns were that the discussion of implicit assumptions and potential drawbacks was limited (W1) and that there were other baselines that could add value to the evaluation (W2). Upon reading the above author responses and the revised manuscript, I believe the additional discussion and analyses add value to the submission.
> > >
> > > I would recommend that two primary assumptions (uniform density, and similar topologies across initial and target configurations) be brought over to the main paper (as oppsed to the appendix).
> > >
> > > As for the baselines, I agree with the authors on the analysis of CPDeform vs Bayesian Optimization (BO). While the performance of both approaches tends to be similar (and one might argue BO might perform even better if more computation were allowed), I'd say the comparison is already unfair. BO requires a lot more compute than CPDeform, and that infact works in favor of CPDeform. Whether there's any interesting connections between the two approaches that may be exploited is an interesting future direction.
> > >
> > > In view of the updated manuscript and the additional results/analyses, I am now leaning to accept this paper. I will update my score to reflect this change of opinion.

---

> > > > ### Author Response · Authors · 2021-11-23
> > > > **Thank you**
> > > >
> > > > Thank you for your detailed and constructive comments. We will follow your recommendation and bring the two primary assumptions to the final version of the main paper. Thank you again for your time.
> > > >
> > > > Best, Authors

---

### Author Response · Authors · 2021-11-18
**General Response: Revision Updated**

We would like to thank the reviewers for their thoughtful and constructive feedback.

We are glad to see that reviewers generally appreciated our paper: the solid experimental results (Reviewer rUod, Ept9, P7JN), rationality and simplicity of optimal transport (Reviewer rUod, Ept9), addressing an important gap in differentiable physics (Reviewer rUod), and writing clarity (Reviewer rUod, Ept9)

We would like to emphasize again that our primary contributions are

- We perform an in-depth study of local optimum issues of differentiable physics solver for initial contact points and switching contact points.
- We propose a principled framework, CPDeform, that integrates optimal transport-based contact discovery into differentiable physics.
- We find that the backbone of CPDeform, the contact point discovery method, can be directly employed by the stand-alone solver to find better initial contact points for single-stage tasks.
- On multi-stage tasks, which are infeasible for the vanilla solver, CPDeform employs a heuristic searching approach to iteratively complete the tasks.

According to your comments, we have provided detailed response to each reviewers' questions and concerns. We have revised our paper to include the following changes:
- We have added discussion on the implicit assumptions and their potential limitations in Appendix G. (Reviewer rUod)
- We have included experimental results with multiple restarts and Bayesian optimization in Table 3 at  Appendix E. (Reviewer rUod)
- We have added the runtime plots in Figure 8 in Appendix F. (Reviewer rUod, Ept9)
- We have discussed in Section 6 that our heuristic is not affected by the choice of material types (rigid, elastic, or elastoplastic). (Reviewer Ept9)
- We have clarified in Section 6 that our optimal transport heuristic does not require an object to be represented and simulated by MPM. (Reviewer Ept9)
- We have added the definitions of $\alpha, \beta$ back from Appendix B in Section 3.1. (Reviewer Ept9)
- We have added Figure 9 in Appendix H to further support the optimality of the heuristic. (Reviewer P7JN)
- We have added descriptions of how controller optimization is done with differentiable physics in Appendix D. (Reviewer P7JN)
- We have clarified that $d(\cdot, \cdot)$ computes the distance between the particle to the closest point on the manipulator in Section 3.2. (Reviewer P7JN)

We hope our responses have convincingly addressed all reviewers’ concerns. We thank all reviewers’ time and efforts again! Please don’t hesitate to let us know of any additional comments on the manuscript or the changes.

Best, Authors

---

### Decision · Program_Chairs · 2022-01-20

**Decision:**

Accept (Spotlight)

**Comment:**

The reviewers are unanimous that this is a strong submission that deserves to be accepted.